# Ivermectin Effect on In-Hospital Mortality and Need for Respiratory Support in COVID-19 Pneumonia: Propensity Score-Matched Retrospective Study

**DOI:** 10.3390/v15051138

**Published:** 2023-05-10

**Authors:** Jara Llenas-García, Alfonso del Pozo, Alberto Talaya, Nuria Roig-Sánchez, Noemí Poveda Ruiz, Carlos Devesa García, Emilio Borrajo Brunete, Inmaculada González Cuello, Ana Lucas Dato, Miriam Navarro, Philip Wikman-Jorgensen

**Affiliations:** 1Internal Medicine Service, Hospital Vega Baja, 03314 Orihuela, Spain; alfonsodelpozo@gmail.com (A.d.P.); igcuello@hotmail.com (I.G.C.);; 2Foundation for the Promotion of Health and Biomedical Research of the Valencia Region (FISABIO), 46020 Valencia, Spain; 3Clinical Medicine Department, Miguel Hernández University, 03202 Elche, Spain; 4Infectious Diseases Unit, Hospital Reina Sofía, 30003 Murcia, Spain; 5Hospital Pharmacy Department, Hospital Vega Baja, 03314 Orihuela, Spain; 6Microbiology Department, Hospital Vega Baja, 03314 Orihuela, Spain; 7Epidemiology Unit, Public Health Centre, 03202 Elche, Spain; 8Internal Medicine Service, Elda General University Hospital, 03600 Elda, Spain

**Keywords:** ivermectin, SARS-CoV-2, COVID-19, *Strongyloides*, hyperinfection syndrome

## Abstract

Introduction. There is negligible evidence on the efficacy of ivermectin for treating COVID-19 pneumonia. This study aimed to assess the efficacy of ivermectin for pre-emptively treating *Strongyloides stercoralis* hyperinfection syndrome in order to reduce mortality and the need for respiratory support in patients hospitalized for COVID-19. Methods. This single-center, observational, retrospective study included patients admitted with COVID-19 pneumonia at Hospital Vega Baja from 23 February 2020 to 14 March 2021. Because strongyloidiasis is endemic to our area, medical criteria support empiric administration of a single, 200 μg/kg dose of ivermectin to prevent *Strongyloides* hyperinfection syndrome. The outcome was a composite of all-cause in-hospital mortality and the need for respiratory support. Results. Of 1167 patients in the cohort, 96 received ivermectin. After propensity score matching, we included 192 patients. The composite outcome of in-hospital mortality or need for respiratory support occurred in 41.7% of the control group (40/96) and 34.4% (33/96) of the ivermectin group. Ivermectin was not associated with the outcome of interest (adjusted odds ratio [aOR] 0.77, 95% confidence interval [CI] 0.35, 1.69; *p* = 0.52). The factors independently associated with this endpoint were oxygen saturation (aOR 0.78, 95% CI 0.68, 0.89, *p* < 0.001) and C-reactive protein at admission (aOR: 1.09, 95% CI 1.03, 1.16, *p* < 0.001). Conclusions. In hospitalized patients with COVID-19 pneumonia, ivermectin at a single dose for pre-emptively treating *Strongyloides stercoralis* is not effective in reducing mortality or the need for respiratory support measures.

## 1. Introduction

COVID-19, a disease caused by the SARS-CoV-2 virus, was first identified in December 2019 in Wuhan, China, and rapidly spread worldwide. According to WHO data, as of March 2023, more than 761 million cases have been reported with more than 6.8 million deaths [1].

The clinical presentation of COVID-19 ranges from asymptomatic to severe illness, and symptoms may change over the course of illness. Symptoms can overlap with those of other viral respiratory illnesses. Severe illness is defined by oxygen saturation of less than 94% at room temperature and at sea level, a ratio of arterial oxygen partial pressure to fractional inspired oxygen (PaO_2_/FiO_2_) less than 300 mmHg, a respiratory rate of more than 30 breaths/min, or lung infiltrates over 50% [2].

When the COVID-19 pandemic emerged in 2020, it quickly became essential to investigate molecules that could be effective against the virus. Drug repurposing can accelerate the identification of effective compounds for clinical use against SARS-CoV-2, with the advantage of pre-existing clinical safety data, lower costs, and an established supply chain [3]. One of the molecules that arose as a potential candidate was ivermectin (IVM).

IVM is an essential medicine, active against a considerable number of helminths and ectoparasites. Moreover, it has antiviral properties, inhibiting importin α/β1 and, thus, blocking protein nuclear importation of HIV-1 and dengue viruses and their subsequent replication [4]. It has also been shown to inhibit flavivirus replication by targeting NS3 helicase activity. In March 2020, Australian researchers demonstrated that IVM also had antiviral activity against SARS-CoV-2 in vitro in Vero cell cultures, virtually reducing the viral load to zero in 48 h [5]. Later, another mechanism was proposed—blocking the spike protein–ACE2 receptor complex [6]. These studies generated considerable enthusiasm about IVM’s potential role for treating and preventing COVID-19, and many clinical trials were started. At the same time, many people started self-treating with the drug, exposing themselves to the risks of off-label use [7]. Even some governments and health professionals embraced its use without clear evidence of clinical benefit.

Controversy has marked IVM as a potential COVID-19 treatment candidate. Initially some trials showed favorable results, and some meta-analyses reported clinical benefit and even reduced mortality [8]. Later, extensive inconsistencies within the trial data appeared; one study was rated as potentially fraudulent, and meta-analyses were retracted or prompted expressions of concern [9]. In July 2021, a Cochrane Review assessed the evidence base for IVM use for preventing and treating COVID-19 in inpatient and outpatient settings, concluding that the evidence was uncertain and based on small studies, few of which were considered high quality [10].

Recently, Bitterman et al. [11] conducted a meta-analysis separating the studies according to whether they took place in an area endemic for *Strongyloides stercoralis*. IVM is the main treatment for strongyloidiasis, a chronic helminth infection with the potential for producing a severe clinical picture called *Strongyloides* hyperinfection syndrome (SHS), a life-threatening condition characterized by massive multiplication of larvae, typically in immunocompromised hosts. The basis for Bitterman et al.’s analysis is that, in endemic areas, the possibility of SHS in patients with COVID-19 treated with corticosteroids, as well as the possible presence of other parasites that are sensitive to IVM, can act as confounders. Their study showed that IVM protected against mortality in endemic areas (risk ratio [RR] 0.25, 95% confidence interval [CI] 0.09–0.70]; *p* = 0.008), but it had no effect in areas of low prevalence (RR 0.84, 95% CI 0.60–1.18; *p* = 0.31).

The health department of Orihuela is, as other regions of Southern Europe [12,13], an endemic area for *Strongyloides stercoralis* [14]; thus, the use of IVM is protocolized in migrants and in native-born patients over 45 years of age who are going to receive immunosuppressants that cannot be delayed until having the *Strongyloides* serology result. This practice, aimed at avoiding SHS, means that a significant proportion of patients with COVID-19 in our department have received IVM.

Our main objective was to evaluate the efficacy of ivermectin for pre-emptively treating *Strongyloides stercoralis* hyperinfection syndrome in order to reduce mortality and the need for respiratory support in hospitalized patients with COVID-19 pneumonia.

## 2. Materials and Methods

### 2.1. Study Design and Population

This single-center, retrospective observational study included all adults admitted to the Vega Baja Hospital (Orihuela, Spain) with a microbiologically confirmed diagnosis of COVID-19 pneumonia between 23 February 2020 and 14 March 2021, thus including patients during the first three pandemic waves that occurred in Spain. Patients with a previous COVID-19 pneumonia admission and patients without microbiological confirmation were excluded. The center is a regional hospital, located in the Vega Baja region of Alicante, Spain, and covers a population of over 170,000 people.

### 2.2. Data Collection

We retrospectively reviewed the medical records of every patient fulfilling inclusion criteria. Study data were collected and managed using REDCap electronic data capture tools, hosted at the Foundation for the Promotion of Health and Biomedical Research of the Valencian Region (FISABIO). We recorded sociodemographic, clinical, microbiological, analytical, and treatment-related variables. The Spanish Society of Infectious Diseases and Clinical Microbiology (SEIMC) score was used to calculate disease severity [15]. SEIMC score is a prediction score based on readily available clinical and laboratory data (age, age-adjusted low saturation of oxygen, neutrophil-to-lymphocyte ratio, estimated glomerular filtration rate, dyspnea, and sex) and it has proven to be a useful tool to predict 30-day mortality probability among hospitalized patients with COVID-19.

Mortality or readmission data at 30 days from discharge were collected by reviewing each patient’s outpatient and inpatient medical records. The primary outcome was a composite of all-cause mortality or the need for ventilatory support. Secondary outcomes were admission in the intensive care unit (ICU), length of hospital stay, strongyloidiasis seroprevalence, and death and readmission rate at 30 days from discharge.

### 2.3. Statistical Analysis

IVM-treated patients were considered cases. Controls were drawn from a larger sample using a propensity score matching procedure. This selection method consisted of fitting a multiple binary logistic regression; the dependent variable was group membership (test/control), and independent variables consisted of the demographic profile, COVID-19 wave, SEIMC score, diagnosis of pneumonia and symptoms on admission, anthropometric measurements (obesity), smoking, systemic diseases, chronic medications, and COVID-19 treatment (corticosteroids/tocilizumab/remdesivir). Using the model obtained, the probability of assignment to the test group was estimated. Next, a hierarchical agglomerative cluster analysis based on the nearest neighbors method was performed. After determination of the final sample, an analysis of homogeneity of groups (test and control) was undertaken using the chi-squared test, Fisher’s exact test, independent t-test, and Mann–Whitney U test to verify that the selection had been successful.

Continuous variables were expressed as the mean and standard deviation (SD) or median and interquartile range (IQR), as appropriate, while categorical variables were expressed as absolute and relative frequencies.

In the inferential analysis, the chi-squared test of independence was used to assess the association between categorical variables (with Fisher’s exact test when needed), and the independent samples *t*-test was used to compare the means of quantitative variables.

Simple binary logistic regression models were fit to study the association between dichotomous outcomes and independent variables. A selection of the most relevant variables (*p* < 0.10) and possible confounders was used to fit the multivariable model and calculate the adjusted odds ratios (aOR). To evaluate the quality of the resulting model, the Hosmer–Lemeshow test, chi-squared goodness of fit, and Nagelkerke R^2^ tests were used.

To assess the outcome length of hospital stay, single and multiple linear models were used. The quality and validity of these models were evaluated using the coefficient of determination (R^2^), variance inflation factors, the Durbin–Watson statistic, and the residual normality test.

The level of significance used in the analyses was 5% (α = 0.05).

### 2.4. Sample Size

To obtain a confidence level of 95% and a statistical power of 80%, on the basis of the difference in mortality observed in other studies of 15.2% in the IVM group compared to 25% in the group of patients not treated with IVM [16], we calculated that a sample size of 190 patients (95 in each arm) was necessary.

### 2.5. Ethical Considerations

The study was conducted in accordance with the World Medical Association’s Code of Ethics for human experiments (Declaration of Helsinki). At all times, national regulations regarding data protection were followed. The study was reviewed and approved by the local ethics research committee (TFM-2021-013). Informed consent was deemed unnecessary.

## 3. Results

From 23 February 2020 to 14 March 2021, there were 96 cases of microbiologically confirmed COVID-19 pneumonia treated with a single dose of IVM 200 µg/kg (43 patients received 18 mg, 37 patients received 15 mg, 13 patients received 12 mg, and one patient each received 16 mg, 17 mg, and 21 mg). From a sample of 1071 patients, 96 controls were selected. Table 1 presents sociodemographic and clinical characteristics of the 192 patients, while Table 2 shows analytical and vital signs.

### 3.1. Primary Outcome

The in-hospital mortality rate was 10.4% (20/192): 11.5% in controls (11/96) and 9.4% (9/96) in IVM-treated patients (Table 3). Need for respiratory support was 35.9% (69/192): 39.6% (38/96) in controls and 32.3% (31/96) in the IVM group. The composite outcome occurred in 38% of participants (73/192): 41.7% (40/96) in the control group and 34.4% (33/96) in the IVM group.

Figure 1 shows the admission duration for IVM-treated cases and controls.

Table 4 shows the results of the univariable and multivariable analysis. IVM use was not associated with the primary composite outcome of in-hospital mortality and/or need for respiratory support (aOR 0.77, 95% CI 0.35, 1.69; *p* = 0.52). The factors independently associated with this endpoint were oxygen saturation (aOR 0.78, 95% CI 0.68, 0.89, *p* < 0.001) and C-reactive protein at admission (aOR 1.09, 95% CI 1.03, 1.16, *p* < 0.001). The results of the goodness-of-fit tests were as follows: Hosmer–Lemeshow test, *p* = 0.14; chi-squared test, *p* < 0.001; percentage of cases correctly classified, 77.5%; Nagelkerke R^2^, 0.32.

IVM efficacy did not differ by gender; the primary composite outcome occurred in 42.1% of men and 28.2% of women receiving IVM (*p* = 0.165).

### 3.2. Secondary Outcomes

Regarding intensive care, 30.2% of patients were admitted to the ICU: 31.3% of controls and 29.2% of cases. Neither the univariable (OR 0.91, 95% CI 0.49, 1.68, *p* = 0.75) nor the multivariable model (aOR 1.04, 95% CI 0.51, 2.11, *p* = 0.92) showed that IVM had any significant effect on the need for ICU admission (Appendix A).

The median length of stay in the overall sample was 9 days (IQR 6–15). This outcome was similar in cases (9 days, IQR 6–15) and controls (8 days, IQR 5–14) (Figure 1), as confirmed statistically in both the univariable (β −0.09, 95% CI −1.09, 1.90, *p* = 0.93) and multivariable (β 0.31, 95% CI −1.31, 1.94, *p* = 0.70) analyses (Appendix A).

*Strongyloides* serology was performed in only 64 patients (33.3%): seven in the control group (7.3%) and 57 in the IVM group (59.4%). There was only one positive case (1.6% positivity rate) in the IVM group.

The 30-day readmission rate was 2.4% overall, with no significant differences between groups (cases 2.3% versus controls; OR 0.98, 95% CI 0.13, 7.10; *p* = 0.98).

Lastly, the 30-day mortality was 11.6%, with similar rates in both groups (cases 10.4% versus controls 12.8%; OR 0.80, 95% CI 0.33, 1.94; *p* = 0.80).

## 4. Discussion

In this propensity score-matched study in an area endemic for strongyloidiasis, we found no benefit of adding a single, 200 µg/kg dose of IVM in patients hospitalized with COVID-19 pneumonia.

IVM has been tested mainly in high-risk patients with mild to moderate disease and, so far, there is no evidence that it prevents COVID-19 progression [17,18,19,20]. A study in Colombian outpatients showed that a 5 day course of ivermectin did not significantly improve the time to resolution of symptoms [20]. The I-TECH study [17] in Malaysia, in which patients were randomized to placebo or IVM 400 µg/kg/day for 5 days during early illness, found that IVM did not prevent progression to severe disease. Another study in the USA [18] also failed to demonstrate the efficacy of IVM (400 µg/kg/day for 3 days) for improving time to recovery in mild to moderate COVID-19. An Italian study [21] showed that higher doses of IVM (600 μg/kg/day for 5 days and 1200 μg/kg for 5 days) were safe but not effective for reducing viral load in adults recently diagnosed with asymptomatic or oligosymptomatic SARS-CoV-2 infection.

Fewer studies have included inpatients. Ahmed et al. reported a faster viral clearance without clear clinical benefit [22]. A larger study in hospitalized patients with a severe form of COVID-19 in Brazil [23] also showed no reduction in the need for supplemental oxygen, ICU admission, invasive ventilation, or death. On the other hand, Mahmud et al. [24] reported that patients treated with IVM plus doxycycline recovered earlier, were less likely to progress to more serious disease, and were more likely to be COVID-19-negative by RT-PCR on day 14. Abd-El-Salam et al. [25] obtained results similar to ours, reporting no differences in length of stay between patients treated with versus without ivermectin.

The most recently updated Cochrane Review [26] concluded that, for outpatients with COVID-19, there is currently low- to high-certainty evidence that IVM has no beneficial effect, whereas, for inpatients, only very low-certainty evidence is available; hence, there is still uncertainty about IVM’s effects for preventing death or clinical worsening or increasing serious adverse events, while there is low-certainty evidence that it has no beneficial effect for clinical improvement, viral clearance, and adverse events. Our study contributes to increasing the evidence on the absence of clinical benefit of IVM in patients hospitalized with COVID-19 pneumonia.

Lack of efficacy may be due to the doses used. Some experts have signaled that the usual doses of IVM 200–400 μg/kg would not achieve a plasma or lung concentration of IVM compatible with the half maximal inhibitory concentration (IC_50_) found in vitro [27], and some researchers advocated for exploring an inhaled route. Studies of IVM use at higher-than-approved doses are needed to evaluate its safety.

Although our hospital is in an endemic area for *Strongyloides stercoralis*, we did not observe the benefit described by Bitterman et al. in their meta-analysis. However, this result may be attributable to the low seroprevalence of *Strongyloides* in our setting.

Adherence to the recommendation of using IVM pre-emptively, before starting immunosuppressive treatment, is low in our context. This is of concern because SHS is a potentially lethal condition and has been described as secondary to COVID-19 treatment [28]. *Strongyloides* serology is not available at our center; therefore, samples are analyzed at a national reference laboratory, entailing a delay of 4–6 weeks before obtaining results. Empiric treatment is, therefore, the best management option. We agree with other authors that strategies to prevent SHS in COVID-19 patients must be urgently implemented [29], either by screening for strongyloidiasis or treating empirically, with the latter constituting a more cost-effective approach [30]. Automated alerts in patients at risk could be of help.

Our study had several limitations. First, it was a retrospective study, carrying the risk of certain biases, including information bias. In addition, the administration of IVM was not randomized but rather recommended in the COVID-19 treatment protocol and, therefore, subject to the discretion of each physician. Moreover, the IVM dose may not have been homogeneous, as it depended on the type of patient; it was generally given to older patients and immigrants, who may have had a worse baseline prognosis than the remainder of the cohort. To try to overcome these limitations, propensity score matching was carried out to try to control for possible confounders. Lastly, this was a single-center study in an area of low endemicity for strongyloidiasis, limiting the applicability of the results.

Although recent evidence has shifted global attention toward novel (and much more expensive) COVID-19 medications, interest in a possible role for cheaper and older compounds remains. To know if IVM is useful in hospitalized patients, a randomized clinical trial with high-dose IVM would have to be conducted in patients hospitalized for COVID-19 pneumonia. Meanwhile, due to the urgent need to find a treatment, an exhaustive and impartial review is necessary in order to integrate the knowledge that exists to date.

## 5. Conclusions

In conclusion, in this study conducted in a region with low endemicity for strongyloidiasis, a single dose of IVM 200 µg/kg showed no benefit for reducing mortality or the need for respiratory support in adults hospitalized for COVID-19 pneumonia.

## Figures and Tables

**Figure 1 viruses-15-01138-f001:**
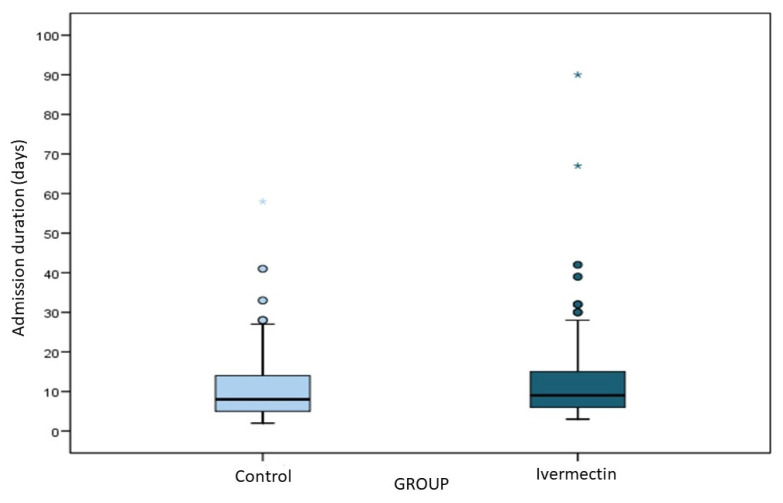
Length of hospital stay in IVM-treated cases and controls. * outliers.

**Table 1 viruses-15-01138-t001:** Sociodemographic and clinical characteristics of sample, according to treatment with ivermectin (IVM).

Variables *		IVM (*n* = 96)	No IVM (*n* = 96)	*p*-Value
Demographics	Women	39 (40.6%)	35 (36.5%)	0.55
	Age in years, mean ± SD	59.8 ± 17.1	60.1 ± 16.2	0.90
	Country of birth (European vs. rest)	48 (50.0%)	51 (53.1%)	0.67
	Race/ethnicity (Black vs. rest)	1 (1.0%)	1 (1.0%)	1.00
Pandemic wave	1st (from 31 January 2020 to 21 June 2020)	5 (5.2%)	2 (2.1%)	0.31
	2nd (from 22 June 2020 to 6 December 2020)	35 (36.5%)	43 (44.8%)	
	3rd (from 7 December 2020 to 14 March 2021)	56 (58.3%)	51 (53.1%)	
Symptomatic		96 (100%)	96 (100%)	1.00
Interval from symptoms to admission, days		6 [4, 9.5]	6.5 [4, 9]	0.79
SEIMC score †		8.8 ± 6.2	8.9 ± 5.7	0.93
SEIMC risk group	Low risk	6 (6.3%)	6 (6.3%)	0.60
	Moderate risk	32 (33.3%)	24 (25.0%)	
	High risk	26 (27.1%)	27 (28.1%)	
	Very high risk	32 (33.3%9	39 (40.6%)	
Comorbidities	Arterial hypertension	39 (40.6%)	38 (39.6%)	0.88
	Diabetes mellitus	21 (21.9%)	18 (18.8%)	0.59
	Hypercholesterolemia	29 (30.2%)	28 (29.2%)	0.87
	Obesity	35 (51.5%)	37 (52.9%)	0.87
	Tobacco use	5 (5.2%)	4 (4.2%)	1.00
	Ischemic heart disease	2 (2.1%)	1 (1%)	1.00
	Chronic obstructive pulmonary disease	4 (4.2%)	8 (8.3%)	0.23
	Asthma	3 (3.1%)	6 (6.3%)	0.50
	Chronic renal failure	7 (7.3%)	8 (8.3%)	0.79
	Cirrhosis	1 (1.0%)	1 (1.0%)	1.00
	Cerebrovascular disease	4 (4.2%)	3 (3.1%)	1.00
	Dementia	1 (1.0%)	1 (1.0%)	1.00
	Neoplasia	10 (10.4%)	14 (14.6%)	0.38
	Intestinal inflammatory disease	5 (5.2%)	8 (8.3%)	0.39
	HIV	0 (0%)	0 (0%)	1.00
	Transplant	1 (1.0%)	0 (0%)	1.00
ICU candidate		82 (89.1%)	83 (90.2%)	0.97
Institutionalized		1 (1.0%)	1 (1.0%)	1.00
Chronic treatment	ACE inhibitors	10 (10.4%)	9 (9.4%)	0.81
	ARA-II	20 (20.8%)	22 (22.9%)	0.73
	Inhaled corticosteroids	7 (7.3%)	11 (11.5%)	0.32
	Inhaled beta-agonists	5 (5.2%)	9 (9.4%)	0.27
	Inhaled anticholinergics	3 (3.1%)	7 (7.3%)	0.19
	Systemic corticosteroids	1 (1.0%)	3 (3.1%)	0.62
	Active chemotherapy	4 (4.2%)	1 (1.0%)	0.37
	Biological treatment	2 (2.1%)	3 (3.1%)	1.00
	Other immunosuppressive treatment	1 (1%)	0 (0%)	1.00
COVID-19 diagnostic test	PCR	74 (77.1%)	79 (82.3%)	0.37
	Antigen test	22 (22.9%)	17 (17.7%)	
Chest X-ray findings	Infiltrate	95 (99.0%)	93 (96.9%)	0.62
	Bilateral infiltrate	86 (90.5%)	85 (91.4%)	0.84
	Pleural effusion	0 (0%)	4 (4.3%)	0.06
Thorax CT	Performed	14 (14.6%)	18 (18.8%)	0.44
	Pulmonary embolism	0 (0%)	1 (5.6%)	
	COVID-19 infiltrates	13 (92.9%)	17 (94.4%)	
Treatment during admission	Corticosteroids	96 (100%)	95 (99.0%)	1.00
	Tocilizumab	5 (5.2%)	2 (2.1%)	0.44
	Remdesivir	36 (37.5%)	30 (31.3%)	0.36

* Frequencies expressed as n (%), mean ± standard deviation or median [interquartile range]. † SEIMC score is a prediction score based on age, age-adjusted low saturation of oxygen, neutrophil-to-lymphocyte ratio, estimated glomerular filtration rate, dyspnea, and sex. It has proven to be a useful tool to predict 30-day mortality probability among hospitalized patients with COVID-19. ACE, angiotensin-converting enzyme, ARA-II, angiotensin II antagonist receptors; ICU, intensive care unit; IQR, interquartile range; PCR, polymerase chain reaction; SD: standard deviation; SEIMC, Spanish Society of Infectious Diseases and Clinical Microbiology.

**Table 2 viruses-15-01138-t002:** Analytical and vital signs, according to treatment with ivermectin (IVM).

Variables *	IVM (*n* = 96)	No IVM (*n* = 96)	*p*-Value
Vital signs at admission			
Systolic blood pressure (mmHg)	136.2 ± 17.1	133.4 ± 18.2	0.29
Diastolic blood pressure (mmHg)	75.8 ± 10.2	74.8 ± 12.4	0.57
Heart rate (beats per minute)	92.0 ± 18.8	94.9 ± 17.9	0.29
Temperature (°C)	37.0 ± 1.0	36.9 ± 1.1	0.55
Oxygen saturation (%)	93.8 ± 4.8	92.8 ± 6.9	0.25
Respiratory rate (breaths per minute)	22 [20, 25]	24 [20–31.5]	0.28 (MW)
Biochemistry at admission			
Glucose (mg/dL)	125.20 ± 43.87	132.70 ± 48.2	0.26
Urea (mg/dL)	37.82 ± 28.96	40.04 ± 23.88	0.56
Creatinine (mg/dL)	1.07 ± 0.54	1.14 ± 0.47	0.37
eGFR (MDRD-4 method) (mL/min/1.73 m^2^)	74.55 ± 26.20	70.75 ± 29.19	0.34
Bilirubin (mg/dL)	0.59 ± 0.37	0.54 ± 0.31	0.40
GOT (UI/L)	64 [50, 98]	75 [56, 105]	0.78 (MW)
GGT (UI/L)	87 [40, 190]	73 [38, 165]	0.17
LDH (UI/L)	362.48 ± 147.31	348.82 ± 155.11	0.68
Creatine kinase (UI/L)	542.57 ± 3927.86	244.33 ± 651.73	0.11
Sodium (mmol/L)	135.08 ± 13.97	136.33 ± 3.49	0.40
Potassium (mmol/L)	3.91 ± 0.54	3.92 ± 0.51	0.87
Ferritin (µg/L)	777.79 ± 920.23	946.03 ± 1230.73	0.36
C-reactive protein (mg/dL)	10.82 ± 7.31	12.28 ± 7.19	0.09
Procalcitonin (ng/mL)	0.38 ± 2.19	0.24 ± 0.79	0.76
Troponin I (pg/mL)	16.32 ± 39.40	31.04 ± 100.65	0.48
Blood cell count and coagulation at admission			
Hemoglobin (g/L)	13.41 ± 1.96	13.55 ± 18.46	0.63
Leukocytes (×10^9^/L)	7.54 ± 3.35	7.25 ± 2.63	0.51
Lymphocytes (×10^9^/L)	1.02 ± 0.66	1.30 ± 2.74	0.32
Neutrophils (×10^9^/L)	6.61 ± 7.13	7.75 ± 11.83	0.42
Platelets (×10^9^/L)	221.91 ± 97.26	224.87 ± 94.31	0.83
aPTT (s)	27.13 ± 16.65	25.59 ± 5.14	0.39
D-dimer (mg/dL)	980 ± 920	1390 ± 2550	0.61
Arterial blood gases at admission			
pH	7.46 ± 0.03	7.45 ± 0.05	0.56
pO_2_ (mmHg)	73.13 ± 29.67	67.06 ± 20.76	0.23
pCO_2_ (mmHg)	33.88 ± 4.40	33.92 ± 5.08	0.97
*Strongyloides* IgG+	1 (1.8%)	0 (0%)	1.00
Parasites in stool	0 (0%)	0 (0%)	1.00

* Frequencies expressed as n (%), mean ± standard deviation, or median [interquartile range]. aPTT: activated partial thromboplastin time; eGFR, estimated glomerular filtration rate; GGT, gamma glutamyl transpeptidase; GOT, glutamic oxaloacetic transaminase; LDH, lactate dehydrogenase; MDRD, Modification of Diet in Renal Disease; MW: Mann–Whitney U test.

**Table 3 viruses-15-01138-t003:** Study outcomes in ivermectin (IVM)-treated patients versus controls.

Outcomes	IVM-Treated Patients (N = 96) *n* (%)	Controls (N = 96) *n* (%)
ICU admission	28 (29.2)	30 (31.3)
Need for respiratory support	31 (32.3)	38 (39.6)
In-hospital deaths	9 (9.4)	11 (11.5)
In *ICU*	6 (6.3)	7 (7.3)
Readmissions at 30 days	2 (2.1)	2 (2.1)
Deaths during readmission	1 (1.0)	1 (1.0)
30 day mortality	10 (10.4)	12 (12.5)

ICU, intensive care unit.

**Table 4 viruses-15-01138-t004:** Logistic regression analysis for primary composite outcome (in-hospital death or need for respiratory support).

	Univariable			Multivariable		
	OR	95% CI	*p*-Value	aOR	95% CI	*p*-Value
Group (IVM vs. control)	0.73	0.41, 1.32	0.30	0.77	0.35, 1.69	0.52
Age	1.01	0.99, 1.03	0.21			0.75
Sex (female vs. male)	0.82	0.45, 1.50	0.51			0.46
Country of birth (non-European vs. European)	0.68	0.38, 1.22	0.68			0.66
Pandemic wave (2nd vs. 1st)	0.98	0.21, 4.67	0.98			
Pandemic wave (3rd vs. 1st)	0.71	0–15, 3.32	0.66			
SEIMC score †	1.05	1.00, 1.11	0.04 *			
Days since symptoms onset	0.91	0.84, 0.99	0.03 *			
Arterial hypertension	1.28	0.71, 2.32	0.41			
Diabetes mellitus	1.99	0.98, 4.04	0.06			0.12
Hypercholesterolemia	1.04	0.55, 1.96	0.92			
Obesity	1.19	0.60, 2.38	0.62			
Smoking	2.11	0.55, 8.14	0.28			
COPD	1.69	0.52, 5.44	0.38			
Asthma	0.45	0.09, 2.23	0.33			
Chronic kidney disease	1.97	0.68, 5.68	0.21			
Cerebrovascular disease	0.26	0.03, 2.22	0.22			
Neoplasia	1.19	0.50, 2.84	0.69			
Inflammatory bowel disease	1.99	0.64, 6.20	0.23			
ICU candidate (no vs. yes)	1.62	0.62, 4.20	0.32			0.92
ACE inhibitors	0.95	0.35, 2.52	0.91			
ARA-II	1.30	0.65, 2.60	0.47			
Inhaled corticosteroids	1.34	0.50, 3.57	0.56			
Inhaled beta-agonists	0.63	0.19, 2.09	0.45			
Inhaled anticholinergics	1.09	0.30, 4.01	0.90			
COVID-19 diagnostic technique (antigen vs. PCR)	1.17	0.57, 2.40	0.67			
Infiltrate at X-ray (unilateral vs. bilateral)	0.32	0.09, 1.14	0.08			0.45
Systolic blood pressure (mmHg)	1.01	0.99, 1.02	0.48			
Diastolic blood pressure (mmHg)	0.98	0.95, 1.00	0.09			0.95
Heart rate (beats per minute)	1.00	0.98, 1.02	>0.99			
Temperature (°C)	1.14	0.86, 1.51	0.36			
O_2_ saturation (%)	0.75	0.67, 0.84	<0.001 ***	0.78	0.68, 0.89	<0.001 ***
Respiratory rate (breaths per minute)	1.06	0.95, 1.16	0.36			
Glucose (mg/dL)	1.00	0.99, 1.01	0.26			
Urea (mg/dL)	1.02	1.01, 1.03	0.006 **			0.09
Creatinine (mg/dL)	2.53	1.32, 4.85	0.005 **			0.24
eGFR (MDRD-4 method) (mL/min/1.73 m^2^)	0.99	0.97, 0.99	0.02 *			0.59
Bilirubin (mg/dL)	0.84	0.35, 2.03	0.70			
GOT (UI/L)	1.02	1.00, 1.05	0.05			
GGT (UI/L)	1.00	0.99, 1.01	0.39			
LDH (UI/L)	1.00	1.00, 1.01	0.09			
Creatinine-kinase (UI/L)	1.00	1.00, 1.01	0.03 *			0.09
Sodium (mmol/L)	1.02	0.97, 1.08	0.49			
Potassium (mmol/L)	0.61	0.33, 1.13	0.12			
Ferritin (µg/L)	1.00	1.00, 1.00	0.74			
C-reactive protein (mg/dL)	1.12	1.06, 1.17	<0.001 ***	1.09	1.03, 1.16	<0.001 ***
Procalcitonin (ng/mL)	9.09	2.10, 39.4	0.003 **			0.36
Troponin I (pg/mL)	1.02	0.99, 1.04	0.12			
Hemoglobin (g/L)	0.98	0.97, 1.00	0.12			
Leucocytes (10^9^ cel/L)	1.13	1.02, 1.25	0.02*			0.34
Lymphocytes (10^9^ cel/L)	1.15	0.87, 1.50	0.33			
Neutrophils (10^9^ cel/L)	1.02	0.99, 1.05	0.27			
Platelets (10^9^ cel/L)	0.99	0.99, 1.01	0.48			
aPTT (s)	1.03	0.98, 1.08	0.23			
D-dimer (mg/dL)	1.68	1.15, 2.44	0.007 **			0.05
Arterial pH	2.20	0.00–21.08	0.87			
Arterial pO_2_ (mmHg)	0.99	0.97, 1.01	0.15			
Arterial pCO_2_ (mmHg)	0.95	0.87, 1.04	0.24			
Tocilizumab	2.24	0.49, 10.3	0.30			0.59
Remdesivir	1.09	0.59, 2.01	0.78			0.27
Corticosteroids	NC	NC	NC			

* *p* < 0.05; ** *p* < 0.01; *** *p* < 0.001. ACE; angiotensin-converting enzyme; aOR: adjusted odds ratio; aPTT, activated partial thromboplastin time; ARA-II, angiotensin II antagonist receptors; CI, confidence interval; COPD, chronic obstructive pulmonary disease, ICU, intensive care unit; LDH, lactate dehydrogenase; OR, odds ratio PCR: polymerase chain reaction; SEIMC, Spanish Society of Infectious Diseases and Clinical Microbiology; NC, not calculable (all but one patient received corticosteroids). † SEIMC score is a prediction score based on age, age-adjusted low saturation of oxygen, neutrophil-to-lymphocyte ratio, estimated glomerular filtration rate, dyspnea, and sex. It has proven to be a useful tool to predict 30-day mortality probability among hospitalized patients with COVID-19.

## Data Availability

The datasets generated and analyzed during the current study are available from the Zenodo repository (doi:10.5281/zenodo.7396689), available at https://zenodo.org/search?page=1&size=20&q=7396689.

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
