# Peer review of "Ivermectin Effect on In-Hospital Mortality and Need for Respiratory Support in COVID-19 Pneumonia: Propensity Score-Matched Retrospective Study"

_viruses, 2023, doi:10.3390/v15051138_

Round 1

Reviewer 1 Report

Comments and Suggestions for Authors

Many thanks to the authors who worked hard on this article. I would like to thank the MDPI editors who invited me to review this manuscript which aimed to evaluate the efficacy of ivermectin for pre-emptively treating Strongyloides stercoralis hyperinfection syndrome in order to reduce mortality and the need for respiratory support in hospitalized patients with COVID pneumonia.

the comments are:

the sample size of 96 is too small to get conclusive results.

I couldn't find the gap of knowledge (what will your study add to the previous studies)

Did authors notice difference among gender in drug efficacy?

Last, if you agree with the following suggestion, I would like to add a timeline image to show the entire process of research.

Reviewer 2 Report

Comments and Suggestions for Authors

The manuscript presented by Jara Llenas-García and et al. entitled Ivermectin effect on in-hospital mortality and need for respiratory support in COVID-19 pneumonia: propensity score matched retrospective study” is of interest, well written, clear, and easy to read. The topic is very interesting and, therefore, in the area of clinical treatment of COVID-19 adding helpful information on how to manage the infection even for future new SARS-CoV2 variants.

I suggest the authors include in the introduction section these two further references concerning the Strongyloides stercoralis in wich IVM is important https://doi.org/10.3390/reports5040047; 10.1016/j.jaccas.2021.04.014
